# Chemical Inhibition of Sterol Biosynthesis

**DOI:** 10.3390/biom14040410

**Published:** 2024-03-28

**Authors:** Eric S. Peeples, Karoly Mirnics, Zeljka Korade

**Affiliations:** 1Department of Pediatrics, University of Nebraska Medical Center, Omaha, NE 68198, USA; zeljka.korade@unmc.edu; 2Child Health Research Institute, Omaha, NE 68198, USA; karoly.mirnics@unmc.edu; 3Division of Neonatology, Children’s Nebraska, Omaha, NE 68114, USA; 4Department of Biochemistry & Molecular Biology, University of Nebraska Medical Center, Omaha, NE 68198, USA; 5Department of Pharmacology & Experimental Neuroscience, University of Nebraska Medical Center, Omaha, NE 68198, USA; 6Munroe-Meyer Institute, University of Nebraska Medical Center, Omaha, NE 68198, USA

**Keywords:** cholesterol, DHCR7, DHCR24, pharmaceutical inhibition of cholesterol synthesis

## Abstract

Cholesterol is an essential molecule of life, and its synthesis can be inhibited by both genetic and nongenetic mechanisms. Hundreds of chemicals that we are exposed to in our daily lives can alter sterol biosynthesis. These also encompass various classes of FDA-approved medications, including (but not limited to) commonly used antipsychotic, antidepressant, antifungal, and cardiovascular medications. These medications can interfere with various enzymes of the post-lanosterol biosynthetic pathway, giving rise to complex biochemical changes throughout the body. The consequences of these short- and long-term homeostatic disruptions are mostly unknown. We performed a comprehensive review of the literature and built a catalogue of chemical agents capable of inhibiting post-lanosterol biosynthesis. This process identified significant gaps in existing knowledge, which fall into two main areas: mechanisms by which sterol biosynthesis is altered and consequences that arise from the inhibitions of the different steps in the sterol biosynthesis pathway. The outcome of our review also reinforced that sterol inhibition is an often-overlooked mechanism that can result in adverse consequences and that there is a need to develop new safety guidelines for the use of (novel and already approved) medications with sterol biosynthesis inhibiting side effects, especially during pregnancy.

## 1. Introduction

Cholesterol is an essential molecule of life, acting as a structural component of all cell membranes as well as an essential regulator of membrane fluidity and cell signaling [1,2,3,4]. The sterol biosynthetic pathway intermediates also serve as precursors for many bioactive molecules, including neurosteroids such as vitamin D [5,6,7]. Pathogenic variants in the sterol biosynthesis pathway that completely eliminate cholesterol production are lethal, while pathogenic variants resulting in significantly diminished cholesterol synthesis give rise to specific clinical syndromes [8,9,10,11]. These syndromes affect the entire body, with systemic dysmorphologies, altered brain development, and subsequent intellectual disability [8].

Cholesterol synthesis can be also inhibited by nongenetic mechanisms, however [12]. Hundreds of chemicals that we are exposed to in our daily lives can alter sterol biosynthesis [13]. Importantly, these include various classes of FDA-approved medications, including (but not limited to) commonly used antipsychotic, antidepressant, antifungal, and cardiovascular medications [14,15,16,17,18,19,20,21]. They can interfere with various enzymes of the post-lanosterol biosynthetic pathway, giving rise to complex biochemical changes throughout the body [22,23].

Cholesterol biosynthesis inhibition can be both beneficial and detrimental, and this depends on the context in which the inhibition occurs [24]. Targeted inhibition by statins is beneficial and saves lives [25], while inhibition as a side effect of psychotropic medications might have unwanted consequences [26]. The impact of these side effects will greatly depend on the stage of life [27,28]. The most robust sterol biosynthesis occurs during the fetal and early postnatal period [29], and developmental interference with this process can lead to lasting consequences that would not be seen in adult exposure [30]. Specifically, the results of a recent review of human clinical data suggested that sterol-inhibiting medications should be considered as teratogens [31].

Chemical inhibition of sterol biosynthesis is a greatly understudied area and is rarely appreciated by researchers or clinicians. To highlight this issue, we performed a comprehensive review of the literature, focusing on the sterol-inhibiting effects and side effects of chemical compounds and prescription medications. The outcome of this review strongly reinforces the notion that sterol inhibition is an often-overlooked mechanism that can result in adverse consequences, and identifies a number of significant gaps in knowledge that should be addressed in future studies.

## 2. Materials and Methods

The literature databases EMBASE and MEDLINE were searched using the search strategies described in Appendix A. Briefly, we sought out studies involving any of the enzymes involved in the Bloch and/or Kandutsch–Russell pathway and any type of medication intervention. We excluded studies that only modified the sterol pathways through genetically altered animals or cells. The titles and abstracts were each reviewed by two authors (E.S.P., Z.K.) to determine relevance for inclusion. If there was ambiguity or disagreement regarding relevance for inclusion at the title/abstract review step, the manuscript was moved along to the full-text review step.

## 3. Results

The initial search returned 518 manuscripts. After initial title and abstract review, 333 manuscripts were excluded, resulting in a total of 185 manuscripts for full text review. Additional relevant manuscripts were identified through review of the reference lists of the included full text review manuscripts. Figure 1 summarizes the inhibitors of each of the key enzymes in the Bloch and Kandutsch–Russell sterol pathways. Additionally, each of those enzymes will be discussed in depth below.

### 3.1. Chemicals That Alter 7-Dehydrocholesterol Reductase (DHCR7) Activity

The *Dhcr7* gene codes for the DHCR7 protein, which is an enzyme responsible for converting 7-dehydrocholesterol (7-DHC) to cholesterol [32,33,34,35]. Mutations in this gene can result in a rare autosomal recessive disorder known as Smith–Lemli–Opitz syndrome (SLOS: OMIM 270400 [36]). The phenotype of this disease can encompass microcephaly with agenesis of corpus callosum, convulsions, micrognathia, upward turned nose, syndactyly, and/or polydactyly [9,37]. Heterozygous carriers of a Dhcr7 mutation will have a slight accumulation of 7-DHC [38], but these smaller changes appear to have no influence on vitality. Several investigators have sought to model DHCR7 inhibition through administration of chemicals that either specifically target DHCR7 production (as in the cases of AY9944 and BM15766) or result in DHCR7 inhibition as an off-target effect. These different chemicals and the data around their use and efficacy are described below.

#### 3.1.1. AY9944

AY9944 is a potent inhibitor of DHCR7 that can mimic the SLOS phenotype [39,40], likely through a combination of direct DHCR7 inhibition in addition to alterations in sonic hedgehog (Shh) signaling, since Shh activation requires covalent linkage to cholesterol [41]. The exact mechanism and effects of changes in Shh signaling are still under investigation [42]. Delivery of AY9944 can model SLOS either through administration during pregnancy to affect the developing fetus or as chronic postnatal administration. Studies assessing administration during pregnancy have found accumulation of 7-DHC, 8-DHC, and trienols in the embryos of the treated dams [43] as well as teratogenicity in rats, reproducing some of the clinical phenotype of SLOS [42].

Chronic AY9944 administration to rats is also used as a model of SLOS and causes progressive irreversible retinal dysfunction and degeneration [44,45]. Xu et al. [46] identified five 7-DHC-derived oxysterols (3β,5α-dihydroxycholest-7-en-6-one (DHCEO), 4α- and 4β-hydroxy-7-DHC, 24-hydroxy-7-DHC, and 7-ketocholesterol) as well as signs of altered bile acid metabolism, membrane dynamics, amino acid catabolism, urea cycle, polyamine synthesis, glucose utilization, and antioxidant mobilization in retinas from AY9944-treated rats. Levels of the branched chain amino acids (BCAAs) were significantly lower in treated retinas compared to controls, while BCAA catabolites were higher, suggesting increased BCAA breakdown. Pipecolate, a proapoptotic catabolite of lysine degradation, was also elevated 3–4-fold in brain and retina and >5-fold in the serum of treated animals compared to controls [46]. Although prolonged administration of AY9944 results in significant elevations of 7-DHC, the 8-DHC concentrations and ratio of 8-DHC:7-DHC do not reach the levels of those seen in humans with SLOS [47].

Although prolonged DHCR7 alterations tend to result in negative outcomes, mimicking SLOS, short-term alterations during periods of stress may be protective. For example, *Dhcr7* mRNA has been found to be upregulated in the leukocytes of those with sepsis who have poor outcome, while inhibiting DHCR7 with AY9944 in zebrafish exposed to LPS decreased lethality [48]. The exact mechanism of this protection is still being evaluated, but the benefits of short-term AY9944 appear to be consistent across several disease models. AY9944 suppresses lipid peroxidation and ferroptosis in human hepatocellular carcinoma Huh-7 cells [49], inhibits vesicular stomatitis virus (VSV) infection in a cell culture model [50], and promotes clearance of zika and other viruses in vitro and in vivo through an interferon-dependent mechanism [51]. These effects can be further supplemented by preventing 7-DHC peroxidation with vitamins E and C (plus sodium selenite) [46].

#### 3.1.2. BM15.766

Another specific competitive inhibitor of DHCR7 is the piperazine derivative BM15.766 [52,53,54], which has been shown to increase 7-DHC and 8-DHC in both the liver and testis [55]. Similar to AY9944, BM15.766 is also teratogenic in rats and replicates some of the SLOS phenotype. Near-term fetuses exposed to maternal administration of BM15.766 between gestational day 1 and 11 demonstrated facial malformations and brain anomalies along the holoprosencephaly spectrum, including pituitary agenesis associated with maternal decreased cholesterol and increased 7-DHC concentrations [56]. Similarly, chronic postnatal administration of BM15.766 to rats starting at the time of weaning resulted in learning deficits that were partially recovered by adding 2% cholesterol to the feeding regimen [54], which not only recovers low cholesterol concentrations but also reduces plasma 7-DHC after BM15.766 treatment [57].

#### 3.1.3. Other DHCR7 Inhibitors

Several in silico and high-throughput in vitro studies have sought to uncover additional chemicals that result in elevated 7-DHC through the inhibition of DHCR7. One such study used the Distributed Structure-Searchable Toxicity (DSSTox) Database Network developed by Environmental Protection Agency to screen environmental molecules that display structures similar to AY9944 and identified the widely used disinfectant benzalkonium chlorides as a potent inhibitor in the cell culture system [19]. When administered in vitro to mouse neurospheres cultured from embryonic neuronal progenitor cells, benzalkonium chloride inhibits DHCR7; however, when administered to pregnant mice, its actions are much more nonspecific, inhibiting multiple sterols (cholesterol, dehydro-cholesterol, zymosterol, desmosterol, lanosterol, lathosterol) in the brain of postnatal day 0 pups [19,20,21].

Other studies have used high-throughput in vitro screening to evaluate over 5000 currently used chemicals to evaluate those that resulted in the off-target effect of 7-DHC elevation. The list of chemicals that resulted in the highest magnitude of 7-DHC change include several antipsychotics and antidepressants as well as the beta-blocker metoprolol [15,16]. Several of these chemicals will be discussed in greater detail below.

Although much of the data on the effects of antipsychotics on 7-DHC have focused on the atypical antipsychotics cariprazine and aripiprazole, the first-generation antipsychotic haloperidol also shows similar effects. Haloperidol treatment of primary neurons and astrocytes inhibited DHCR7 as well as other enzymes in the post-lanosterol pathway [58]. Similarly, in vivo treatment of adult rats resulted in a dose-dependent increase in the brain concentrations of 7-DHC [59].

Cariprazine, an atypical antipsychotic, is one of the most potent 7-DHC-elevating medications, demonstrating similar increases in 7-DHC compared to AY9944 when administered to Neuro2a or human fibroblast cells [60]. Its effects have been tested in several different models, showing similar significant elevations of 7-DHC in vitro in Neuro2a cells infected by VSV [50] as well as in multiple organs, including the fetal brain in vivo after maternal in utero exposure [61].

Aripiprazole, another atypical antipsychotic, has been associated with increased 7-DHC and 8-DHC in human blood samples [59]. Both cariprazine and aripiprazole significantly altered cholesterol biosynthesis precursor profiles in highly proliferative NPCs and early post-mitotic neurons differentiated from human-induced pluripotent stem cells [62]. Additionally, aripiprazole increased 7-DHC in airway and bronchial epithelial cells and potentiated ozone-induced cytokine release in a donor sex-specific manner [63]. Both cariprazine and aripiprazole share the common metabolite 2,3-DCPP which is a potent inhibitor of DHCR7 as well, likely due to the shared dichlorophyenil-piperazine substructure in aripiprazole, cariprazine, and 2,3-DCPP [60].

Trazodone, a commonly prescribed antidepressant, is also a potent DHCR7 inhibitor. Trazodone inhibition of DHCR7 increases 7-DHC and 7-DHD and decreases desmosterol, though its effects on decreasing cholesterol have not been consistently demonstrated. Where the other medications discussed above (haloperidol, cariprazine, aripiprazole) have more complex sterol disruption profiles—with many inhibiting other enzymes at higher doses and increasing zymosterol, zymostenol, lathosterol, DHL, 8-DHD, and 8-DHC—trazodone is mostly a DHCR7 inhibitor [58]. Similar to aripiprazole, trazodone has also been associated with increased 7-DHC and 8-DHC in human blood samples [59,64], and the presence of trazodone in perimortem toxicology screens was strongly associated with observed differences in 7-DHC and desmosterol concentrations in postmortem brain samples [65]. These human studies are further supported by many studies where the levels of 7-DHC were measured after in vitro [50] or in vivo [58,59,64,66,67] exposure to trazodone.

Several other medication classes have also been found to inhibit DHCR7, including in utero exposure to fentanyl [68]. After early fentanyl exposure during pregnancy, a series of ten patients demonstrated clinical phenotypes mimicking SLOS, including microcephaly, bilateral 2,3-toe syndactyly, single palmar creases, and cleft palate. Initial screening showed elevated 7-DHC and 8-DHC, and after one month, follow-up showed normal sterol values. No genetic cause was identified. This report is unique because it is the first to raise concern potentially correlating in utero exposure to fentanyl to neonatal 7-DHC concentrations.

Additionally, administration of the beta-blocker metoprolol to pregnant mice also inhibits DHCR7 in the brains of newborn pups, increasing 7-DHC and decreasing desmosterol [69]. The DHCR7 inhibition by metoprolol is thought to be part of the mechanism regarding why it has some inhibitory effect on VSV infection in cell culture [50]. The newer beta-blocker nebivolol is 10 times more potent in inhibiting DHCR7 in both HepG2 and Neuro2a cell cultures than metoprolol [69].

Lastly, truncated adenomatous polyposis coli selective inhibitors (TASIN), LK-980, and UV radiation may alter DHCR7 activity. TASINs were found to inhibit EBP, DHCR7, and DHCR24, leading to colorectal cancer cell death [70]. LK-980 ((4-Phenethylpiperazin-1-yl)-1-(pyridine-3-yl) ethanol) was synthesized and tested in HepG2 cells and was a specific inhibitor of DHCR7 with partial inhibition of other enzymes [71]. UV radiation decreases DHCR7 protein level in skin cells (primarily keratinocytes), allowing more 7-DHC to be converted to vitamin D [72]. Keratinocytes have an active sterol synthesis pathway and both DHCR7 and DHCR24 are important in normal keratinocytes physiology [73].

### 3.2. Chemicals That Alter 24-Dehydrocholesterol Reductase (DHCR24) Activity

The *Dhcr24* gene codes for the DHCR24 protein, which is a nearly ubiquitous enzyme in sterol synthesis, as it not only catalyzes the conversion of desmosterol to cholesterol but can also convert several of the sterol intermediates in the Bloch pathway over to the corresponding intermediates in the Kandutsch–Russell pathway [74]. DHCR24 deficiency is the underlying mechanism for desmosterolosis (OMIM 602398 [75]), a genetic syndrome characterized by elevated desmosterol concentrations and multiple congenital anomalies [76,77].

DHCR24 is also known as SELective Alzheimer’s Disease INdicator-1 (seladin-1) as it was shown to be downregulated in Alzheimer’s disease (AD) [78]. In addition to AD, however, DHCR24 is also altered in oncogenic and oxidative stress [79], hepatitis C virus (HCV) infections [80], and the development of foam cells [81]. Specifically, *Dhcr24* inhibition and the accumulation of desmosterol might have antiapoptotic activity [82] and anti-inflammatory effects [81] and may decrease hepatitis C viral infection [80,83]. DHCR24 is also a key enzyme in bone development, as DHCR24 KO metatarsal cultures have poor growth due to the absence of proliferating chondrocytes in the growth plate and abnormal hypertrophy of prehypertrophic chondrocytes [84]. Ultimately, the role of DHCR24 in AD remains controversial, but it has also been postulated to be a possible medication target for HCV infections [80] and arteriosclerosis [85]. 

Desmosterol is endogenous agonist for the liver X receptor (LXR) which is a master regulator of lipid metabolism with potential role in inflammation, atherosclerosis, cancer, diabetes mellitus, multiple sclerosis, nonalcoholic steatohepatitis, and viral infections [86]. Many studies have shown that lipid and cholesterol metabolism are critical in the development and progression of different types of tumors, and the overexpression of cholesterol synthesis genes is associated with resistance to conventional antitumor pharmaceuticals [87,88]. As such, many ongoing research studies are exploring the combination of inhibiting sterol enzymes such as DHCR24 with other antitumor treatments.

#### 3.2.1. U18666A

Molecular dynamics simulations of DHCR24, desmosterol, FAD, and U18666A showed that U18666A interacts with flavin adenine dinucleotide (FAD) by forming three hydrogen bonds with the Lys292, Lys367, and Gly438 of DHCR24. U18666A induces secondary structural changes in the interaction of DHCR24, FAD, and desmosterol, thereby blocking DHCR24 activity through an allosteric-inhibiting mechanism [89]. Administration of U18666A has been shown to reproduce the effects of DHCR24 knockout in bone development [84], and has been tested in various systems, as described throughout the rest of this section.

Given the early association with AD, it is not surprising that much of the research using U18666A to date has assessed changes in brain sterol synthesis. For example, intracerebral injection of rats with U18666A for 14–21 days led to decreased cholesterol and increased desmosterol, amyloid beta accumulation, decreased neuron-specific enolase, decreased p-Akt and p-GSK-3beta, and Morris water maze cognitive impairment [90]. Additionally, in a permanent middle cerebral artery occlusion model, DHCR24-HET and WT mice treated with U18666A both had increased ischemic lesions at 48 h after injury [91]. The authors concluded that the DHCR24 enzyme has a neuroprotective role against cell death after ischemia, which was most likely through altered association of other proteins with the lipid rafts. In a cell culture system, U18666A also prevented the cellular accumulation of disease-associated isoforms of prion proteins, though this finding was not reproduced in mice [92].

As noted above, DHCR24 is thought to play a significant role in cancer, so U18666A has also been tested in cancer models. As an example, simultaneous treatment of melanoma cell lines with FASN and DHCR24 inhibitors (PLX4032 and U1866A) increased number of apoptotic cells [93]. Additionally, DHCR24 is coexpressed with parvalbumin in cochlear hair cells in rat organs of Corti and is upregulated in response to cisplatin-induced injury. The inhibition of DHCR24 with U18666A increased the sensitivity of hair cells to cisplatin-induced toxicity [94]. One infectious risk factor for the development of various cancers is chronic HCV infection. HCV in human hepatocytes induces expression of DHCR24 in vitro, and a similar effect was seen in human hepatocytes of chimeric mouse liver. Both treatment with U18666A and DHCR24 siRNA suppressed HCV infection in cell lines [80].

#### 3.2.2. SH42

In 2017, Müller et al. sought to synthesize a novel *Dhcr24* inhibitor using the structures of several known sterol inhibitors (e.g., U18666A, MGI-21, etc.) as a basis. Of the novel compounds that were developed and tested in mice, compound 27 (coded as SH-42) was chosen due to its selectivity, activity, and lack of cytotoxicity [95]. A subsequent study demonstrated that SH42 led to accumulation of desmosterol, increased biosynthesis of polyunsaturated fatty acids, and production of anti-inflammatory mediators [96,97]. The study concluded that DHCR24 is involved in inflammation because desmosterol influences PUFA synthesis—PUFAs are substrates for production of lipid mediators in onset and offset of inflammation—in macrophages through LXR binding.

#### 3.2.3. Other DHCR24 Inhibitors

Screening a collection of FDA-approved medications identified 49 compounds that inhibit DHCR24 and elevate desmosterol levels in vitro in Neuro2a cells, including the tyrosine kinase inhibitors imatinib, ponatinib, and masitinib [15]. An in silico screening used the DrugBank database and molecular dynamics simulation analysis which provided four potential DHCR24 inhibitor candidates: irbesartan, risperidone, tolvaptan, and conivaptan. All four significantly lowered cholesterol in HepG2 cells [98]. Both the types of cells used and the methods of sterol detection were different between these two studies, which are likely the reasons for the lack of overlap in candidate chemicals between the two studies.

Additionally, other studies have demonstrated changes to DHCR24 with chemicals that were not discovered in either of the two screening studies above. The most studied of these has been amiodarone, which was shown to inhibit both EBP and DHCR24, as confirmed by sterol measurements in liver and kidney cell lines [18,99]. Further supporting the in vitro studies, patient serum samples containing detectable amounts of amiodarone also had elevated levels of the sterol precursors zymosterol, 8-DHC, and desmosterol [99]. A separate clinical study performed sterol analysis in 236 cardiac patients (126 with and 110 without amiodarone treatment) and showed that amiodarone administration was accompanied by a robust increase in serum desmosterol levels independently of gender, age, body mass index, cardiac and other diseases, and the use of statins [18]. The patient samples taken before and after initiation of amiodarone therapy showed a systematic increase in desmosterol upon amiodarone administration, suggesting a direct causal link between amiodarone administration and desmosterol accumulation.

Post-transcriptional DHCR24 enzyme activity regulation can also be accomplished through phosphorylation [100] or feedback inhibition from oxysterols [101,102]. Specifically, phosphorylation of the residues T110, Y299, and Y507 are key in the regulation of DHCR24 activity. Inhibition of protein kinase C also greatly decreases DHCR24 activity, though not through the phosphorylation of the known residue targets [100]. Additionally, the oxysterol 24(S),25-epoxycholesterol (24,25EC), which is structurally similar to desmosterol, directly inhibits DHCR24 activity [101]. This oxysterol inhibited enzymatic activity of DHCR24 without affecting its protein level and resulted in increased level of desmosterol and decreased levels of cholesterol [102].

As mentioned in the DHCR7 section, TASINs target EBP, DHCR7, and DHCR24 enzymes in human colonic epithelial cells and carcinogenic HCEC [70,103]. These findings were further supported by a study to develop small molecules toxic to colorectal cancer cells with cancer-causing mutations, in which the researchers identified several different TASIN compounds. They noted that one specific TASIN, TASIN-1, inhibited EBP, DHCR7, and DHCR24, but in their study, the toxic effects of TASIN were exclusively dependent on EBP inhibition in their particular DLD-1 cell line [70].

Several other chemicals have been shown in single studies to potentially inhibit DHCR24 but have not yet been reproduced. These include studies on the phytochemicals in Cannabis sativa [104] and the chemical genkwadaphnin, extracted from the flower buds of Daphne genkwa [105]. The synthetic gamma secretase modulator E2012 also inhibited DHCR24 and led to development of cataract in rats, with increased desmosterol and decreases cholesterol in lens, liver, and plasma [106]. LK-980 (4-Phenethylpiperazin-1-yl)-1-(pyridine-3-yl)ethanol) primarily inhibited DHCR7 in HepG2 cells, but also inhibited DHCR14, DHCR24, and SC5D to a lesser extent [71]. Lastly, metabolic and culture conditions may be important for DHCR24 expression, as high glucose added to medium was shown to decrease *Dhcr24* mRNA in one specific cell type [107]; however, it is unclear if this mRNA change translates into changes in sterol concentrations.

### 3.3. Chemicals That Alter Lanosterol Synthase (LSS) Activity

The primary function of LSS is to convert 2,3-oxidosqualene into lanosterol [108,109]. *LSS* mutations have been associated with cataracts (OMIM 600909 [110]), hypotrichosis (OMIM 618275 [111]) in affected children, and a phenotype referred to as alopecia-intellectual disability syndrome-4 (OMIM 618840 [112]) [113,114]. Although it is a less-studied entity, chemical-induced modeling of LSS deficiency may be particularly relevant because genetic models to date have either relied on additional mutations such as Fdft1 to produce the phenotype [115] or are lethal to the embryos [116]. Additionally, LSS inhibitors have clinical relevance as potential antifungal and/or cholesterol-lowering compounds.

Several studies have screened and prioritized potential LSS inhibitory compounds, including extracts from foods, in different model systems [117,118,119]. These include the experimental compound BIBB-515, which induced 24(S),25 epoxycholesterol and resulted in decreased human rhinovirus replication [120] and sensitized chronic lymphocytic leukemia cells to chemotherapeutics [121]. Diverting sterols into the “shunt” pathway starting with 24(S),25 epoxycholesterol by MM0299 administration also inhibited growth of glioma stem-like cells [122]. Lastly, essential oils behaved similar to the positive LSS inhibitor control Ro 48-8071 in decreasing intracellular lipid levels and cholesterol synthesis and were, thus, hypothesized to function as natural compounds against atherosclerogenesis [123].

As mentioned above, LSS inhibitors have also been shown to have antifungal and antiprotozoan properties, including epohelmins A and B [124]. Due to their focus on microorganisms, many of the studies assessing LSS inhibition are solely in vitro analyses. Although many of these studies suggested LSS inhibition, it should be noted that the concentrations of chemical administered were widely variable: from 500 μM eremanthine [119] to screening several compounds at 20–50 μM [118] and 0.25–0.5 μM diacylglycerols [125].

### 3.4. Chemicals That Alter Sterol-C4-Methyl Oxidase-like (SC4MOL) Activity

SC4MOL deficiency (OMIM 616834 [126]) can result in microcephaly, congenital cataracts, and psoriasiform dermatitis [127]. SC4MOL is highly conserved, and several of the studies targeting its activity have, therefore, been studied in fungal species either to test their antifungal effects or as a model species. In either case, further validation in mammalian cells or animal models would be required before conclusions can be formed about their potential for modeling SC4MOL deficiency.

Seeking additional effective antifungal therapies, one group demonstrated that administering diazaborines to Saccharomyces cerevisiae leads to accumulation of ERG25 substrate, 4,4-dimethylzymosterol, and depletion of zymosterol and ergosterol [128]. Similarly, eugenol—one of the main ingredients in Syzygium aromaticum (cloves) and an inhibitor of SC4MOL—may inhibit mycelia in Rhizoctonia solani, a cause of rice sheath blight. Eugenol downregulated expression of C-4 methyl sterol oxidase, inhibited synthesis of ergosterol, and led to increased oxidative stress [129].

Similar to LSS inhibition, inhibiting SC4MOL may aid in sensitizing cancer cells to chemotherapy. Although they used siRNA and shRNA rather than medication treatment, Sukhanova et al. demonstrated that SC4MOL inactivation sensitized A431, SCC61, SCC68, and PC9 tumor cells to erlotinib, an EGFR kinase inhibitor [130]. The mechanistic theory for this is that specific sterol metabolites (specifically FF-MAS and T-MAS) interfere with endosomal trafficking of EGRF and limit EGFR stability.

Overall, inhibiting enzymes in the cholesterol synthesis pathways tends to result in cellular dysfunction and/or death. One exception to this, that will be explored further in the section below, is that genetic inhibition of SC4MOL increases formation of oligodendrocytes in mouse oligodendrocyte precursor cell culture [131]. This group used the small molecule inhibitor CW4142 which led to increased accumulation of 8,9-unsaturated sterols.

### 3.5. Chemicals That Alter Emopamil-Binding Protein (EBP) Activity

Mutations in the *EBP* gene [132] have been associated with two different primary phenotypes. The first is X-linked dominant chondrodysplasia punctata (OMIM 302960 [133]), in which affected females demonstrate elevated levels of 8-dehydrocholesterol and 8(9)-cholestenol, skeletal dysplasia, and hyperkeratotic skin changes [134]. An X-linked recessive mutation has also been documented in males and has been termed male EBP disorder with neurologic defects (MEND, OMIM 300960 [135]). MEND results in a highly variable phenotype that may include intellectual disabilities, spine and digit abnormalities, cataracts, and/or dermatologic changes.

Long et al. described the crystal structure of EBP, demonstrating a fold involving five transmembrane helix that created a membrane cavity binding site that can accommodate multiple different pharmacological ligands [136]. The compounds that bind with EBP tend to have a positively charged amine group. Several molecules have been identified that have moderate or high affinity for EBP, including tamoxifen, clomiphene, amiodarone, haloperidol, and opipramol [137]. Novel chemicals have also been developed, such as the 2- or 2,6-disubstituted (CH3, CH 3O, Cl, F) cis- and trans-4-(4-aryl)cyclohexyl-1-(2-pyridyl) piperazines, that also inhibit EBP [138]. The most selective 2,6-dimethoxy derivative (cis-33) demonstrated the highest potency (EC 50 = 12.9 μM) and efficacy (70%) in inhibiting proliferation of the human prostate cancer PC-3 cell line. Lastly, in addition to inhibition of DHCR7 and DHCR24, TASINs have also been shown to inhibit EBP [70]. In fact, while TASINs inhibit DHCR7 and DHCR24 in DLD1 and HT29 cell lines, only antitumor effects of TASINs—and especially those of TASIN-1 [139]—appear to act through modulation of EBP expression and not DHCR7 or DHCR24 [70].

As noted in the SC4MOL section, inhibition of some of the sterol synthesis enzymes has been associated with enhanced oligodendrocyte formation and myelination. This effect has been most consistently documented with EBP inhibition. One screening study of a structurally diverse library of 10,000 small molecules found that many of those molecules that effectively promoted oligodendrocyte formation also inhibited the cholesterol synthesis enzymes CYP51, TM7SF2, or EBP [140]. These molecules included the potent EBP-inhibitor, CW3388, which enhanced oligodendrocyte formation in mouse cell cultures.

The mechanism by which EBP inhibition promotes oligodendrocytes is not yet fully known, but the process appears to rely on the accumulation of 8,9-unsaturated sterols [141,142]. In the first days in vitro, oligodendrocyte precursor cell maturation involves high levels of accumulation of zymostenol and zymosterol [143], and inhibition of EBP using peripheral injection of tamoxifen and clemastine in a mouse model of multiple sclerosis resulted in significantly improved myelin repair [143]. Further screening demonstrated that inhibiting delta (14)-sterol reductase with the chemical U-73343 provided similar enhancement of oligodendrocyte formation, also likely through the accumulation of 8,9-unsaturated sterols [144]. The pro-oligodendrocyte effect does not fully rely on 8,9-unsaturated sterols, however, as similar effects were also seen with LSS inhibition and exogenous administration of 24,25-epoxycholesterol [145]. Overall, these studies demonstrate promise for sterol inhibition in enhancing myelination, but future studies will need to attempt to answer the question of how to provide promyelination without suffering the negative effects of sterol inhibition described throughout the rest of this review.

### 3.6. Chemicals That Alter Cytochrome P450 Family 51 Subfamily A Member 1 (CYP51A1) Activity

CYP51A1 is involved in the removal of the 14⍺-lanosterol methyl group in the conversion of lanosterol to 4,4-dimethylcholesta-8(9),14,24-trien-3β-ol, and is, therefore, also referred to as lanosterol 14⍺-demethylase [146]. Although a few studies have sought to alter CYP51A1 in mammalian cholesterol synthesis [147], the vast majority of CYP51A1 research has focused on its vital role in the synthesis of ergosterol in nonhuman species such as fungi [148,149]. Specifically, azole CYP51A1 inhibitors have been developed and are widely used in medicine and agriculture to treat fungal infections [150,151]. The accumulation of these chemicals in soil and waterways could pose risks to the environment [152] and is thought to disrupt sex hormone synthesis and signaling involved in fetal development [153] and pubertal maturation [154]. A full discussion of the mechanisms of these risks is outside the scope of this current review, however.

### 3.7. Chemicals That Alter Sterol-C5-Desaturase (SC5D) Activity

SC5D deficiency can result in lathosterolosis, a clinical syndrome with a complex phenotype involving multiple congenital anomalies, intellectual disabilities, and liver failure [155]. Other than the observation that SC5D is a minor target of the chemical LK-980 [71], very little is known about how many or which medications may affect SC5D activity.

## 4. Discussion

Cholesterol is an important and incredibly complex small molecule—a Janus-like chemical that is simultaneously both life-sustaining and potentially life-threatening [156,157,158,159,160]. Cholesterol and its intermediates are essential structural components of all cell membranes, serve as precursors for hundreds of bioactive compounds, regulate cell division and membrane raft assembly, and serve as molecular underpinnings for all specialized functional processes of the body [161,162,163,164]. Due to the highly conserved biosynthesis pathways across vertebrate species, we can study its biosynthesis in animal models with both genetic and chemical tools [165]. While the biochemical pathways of sterol biosynthesis have been studied for decades (and we have a reasonably good understanding of them), the consequences of sterol biosynthesis disruption by xenobiotic agents are extremely understudied. In particular, the study of the adverse consequences of various chemicals on the fetal sterol biosynthesis continues to be full of opportunities to address critical missing knowledge.

There are hundreds of chemicals that can inhibit various parts of the sterol biosynthesis pathways. Many of these chemicals are FDA-approved medications that are prescribed hundreds of millions of times each year in the USA alone. Through the review of the literature, we identified many areas with gaps of knowledge that could and should be addressed in future studies. These gaps can be largely divided into two main areas: mechanisms by which sterol biosynthesis is altered and consequences that arise from the inhibitions of the various steps of the sterol biosynthesis pathway.

Currently, we have very limited data of the mechanisms by which the various prescription medications and nonprescription chemicals inhibit the sterol biosynthetic processes. For example, do the various chemicals act as direct inhibitors of enzymes through competitive binding for their substrates, or do they influence enzyme levels via secondary transcriptional mechanisms? Additionally, the mechanistic experiments have historically focused on the effects on modulating cholesterol levels, with much less attention paid to the biological roles of the accumulating or decreasing precursor or metabolite levels. Yet, we know that many of the precursors and metabolites have important biological activities that can be beneficial (e.g., decreasing viral load [50,51]) or detrimental (e.g., disrupting critical patterning through the sonic hedgehog pathway [166]). These precursors and metabolites modulate immune function, growth, synapse formation, and many other developmental and homeostatic processes.

Many sterol biosynthesis inhibiting chemicals share a degree of structural similarities (Figure 2), which could be responsible for the inhibition of sterol biosynthesis enzymes, particularly DHCR7. For example, 2,3-dichlorophenyl piperazine (2,3-DCPP) is a substructure and metabolite of both aripiprazole and cariprazine, and by itself it is a DHCR7-inhibiting compound [60]. In addition, 4-chlorophenyl-4-hydroxypiperidine (CPHP) is a structural component of both haloperidol and penfluridol, two antipsychotics that also elevate 7-DHC. Although 2,3-DCPP and CPHP are distinct chemical structures, a side-by-side comparison between the structures of 2,3-DCPP and CPHP shows that both have terminal modified phenyl rings linked to either a piperidine or piperazine ring. Such relationships between sterol biosynthesis inhibition and common chemical substructures should be further investigated by computational approaches that involve systematic fragmentation of molecules (e.g., molBLOCKS tool), followed by enrichment analysis [167]. The identification of these sterol-biosynthesis-inhibiting substructures will be important for knowledge-based drug development, especially for evaluation of medications that can be safely used during pregnancy.

Human studies clearly indicate that post-lanosterol biosynthesis inhibitors act as teratogens [31], yet these unwanted consequences of sterol inhibition are still mostly unknown. This is particularly throughout the critical time windows of exposure—such as during fetal development—that give rise to the most consequential disruptions of development. Similarly, we do not yet understand the vulnerability of the different tissues and what metabolite level thresholds result in toxicity and long-lasting phenotypical changes. In addition, due to the all-encompassing metabolic role of this pathway, phenotypes will likely vary significantly in which organs and biological processes are involved, impairing the ability to developed focused, targeted phenotypic detection.

Outside of teratogenicity, however, there is a paucity of data directly linking the sterol effects of the medications in this review to their clinical effects. For example, the central nervous system side effects of beta-blockers have always been assumed to be brain adrenergic receptor-mediated (and related to their main mechanism of action), although there is very sparse experimental evidence that proves this assumption [168]. Untangling mechanisms of both the main action and side effects is an extremely challenging task, especially if different mechanisms of action occur simultaneously. A recent study evaluated interactions between antipsychotics and medications used in the treatment of cardiovascular disease (CVD) [26]. The highest number of interactions was recorded among beta-blockers and antipsychotics [169]. Although this study only examined cardiovascular side effects, it is notable that the combinations that reported the most common adverse outcomes included the medications identified above as having 7-DHC elevating side effects—including metoprolol and nebivolol. Despite these existing concerns, however, beta-blockers are still often prescribed in addition to standard antipsychotic regimes to treat CVD or extrapyramidal symptoms such as akathisia [169,170].

The highest toxic intermediates will most certainly give rise to complex dysmorphologies and intellectual disabilities. Are lower levels of the same intermediates safe, however, or do they give rise to more nuanced changes (without dysmorphology) that manifest later in life as mild to moderate functional impartments such as autism or learning challenges? For example, the grave disruption of post-lanosterol biosynthesis in SLOS gives rise to both congenital malformations and intellectual disability, with 75% of patients on the autism spectrum [171,172,173]. What remains unknown is whether a lesser magnitude of 7-DHC elevation could lead to only a milder behavioral phenotype without dysmorphology. To add additional complexity, there are several highly oxidizable molecules in this pathway (such as 7-DHD and 8-DHC) that could alter the oxidative milieu in acute periods of stress, yet we have not yet even identified the oxysterols that arise by their autooxidation [174]. Additionally, polypharmacy is a significant challenge throughout clinical medicine [175], and we lack adequate knowledge around the summative or potentiating effects of simultaneously utilizing multiple medications with sterol-biosynthesis-inhibiting side effects. Lastly, many psychotropic medications inhibit the post-lanosterol pathway, but can we safely assume that sterol inhibition is merely a side effect of the medications, or could the sterol biosynthesis effects actually contribute to their therapeutic efficacy?

The method of addressing all these remaining questions will depend on the various in vivo and in vitro models described above to thoroughly investigate sterol inhibition. Luckily, there are a plethora of advanced analytical methods by which the meaningful results can be obtained, but the outcome of these experiments (and the reproducibility of the findings) will greatly depend on the quality of the experimental design. Investigators should pay great attention to the dose of the inhibitor, duration and frequency of exposure, developmental age, genotype, assay strengths and limitations, statistical approaches, power of studies, and many other factors. Additionally, biochemists and bench researchers should work to partner with practicing clinicians to both increase the translatability of the research being performed and to help clinicians develop a greater appreciation of the spectrum of commonly used prescription medications that affect sterol biosynthesis, with yet-unknown long-term clinical consequences. A study that can provide a good cautionary example in this context is the newly identified fetal fentanyl syndrome, that presumably arises from DHCR7 inhibition [176].

## 5. Conclusions

Hundreds of chemicals can inhibit sterol biosynthesis, acting through different enzymes and various molecular mechanisms. This area remains greatly understudied both in the context of mechanistic molecular events and consequences on human health. We must develop models and procedures to study these processes and extensively utilize them to develop safety guidelines in this area, especially during pregnancy. Novel chemicals (and particularly medications awaiting FDA approval) should undergo a comprehensive battery of testing for sterol-inhibiting effects, and a significant number of already approved prescription medications should be re-evaluated in the same context.

## Figures and Tables

**Figure 1 biomolecules-14-00410-f001:**
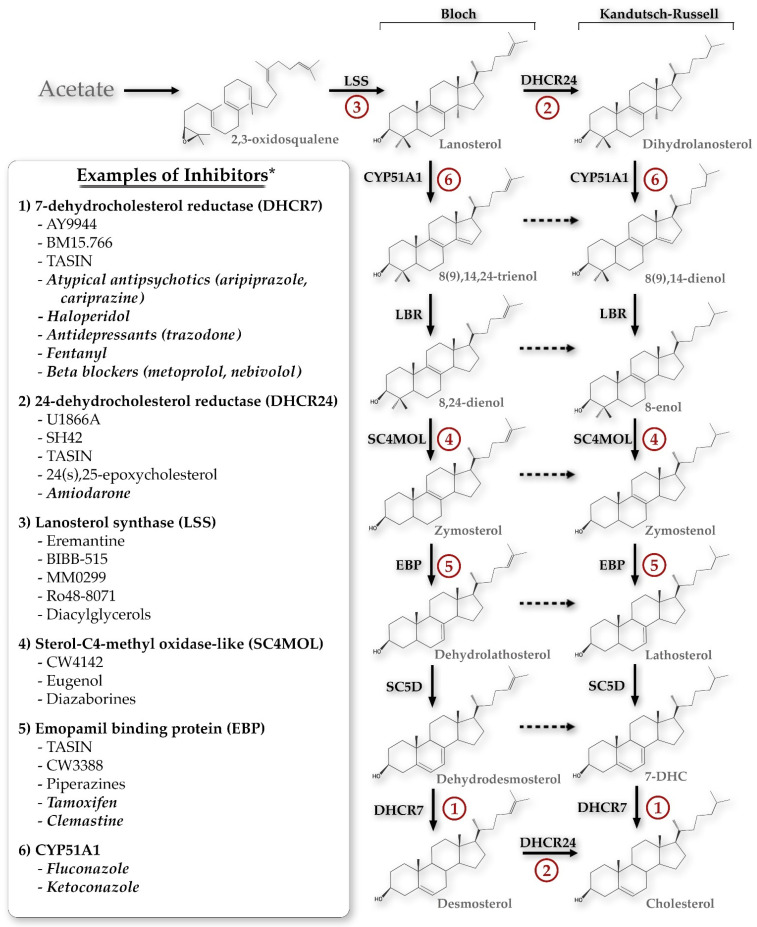
The primary sterol synthesis pathways discussed in this review, demonstrating potential chemical inhibitors for each of the key enzyme steps. * This list is not exhaustive but is meant to demonstrate the breadth of compounds potentially inhibiting each step, including both specific inhibitors and FDA-approved medications (the latter denoted by italics). CYP51A1, cytochrome P450 family 51 subfamily A member 1; LBR, lamin B receptor; SC5D, sterol-C5-desaturase; 7-DHC, 7-dehydrocholesterol; TASIN, truncated adenomatous polyposis coli selective inhibitors.

**Figure 2 biomolecules-14-00410-f002:**
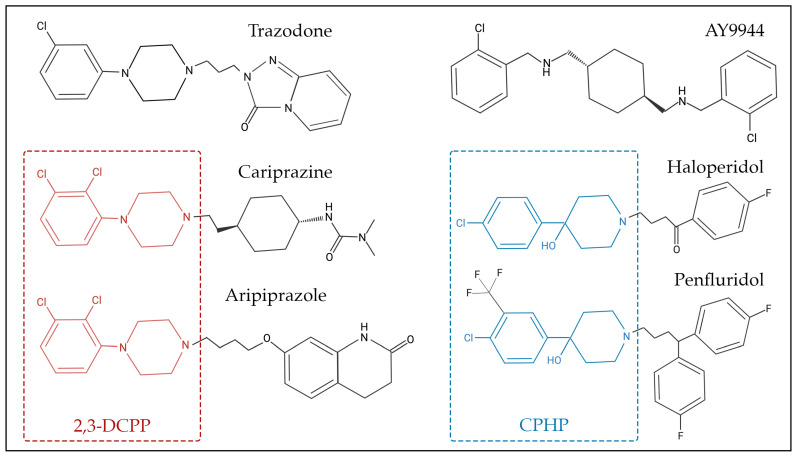
Chemical structure of selected 7-dehydrocholesterol reductase (DHCR7) inhibitors. Some of these compounds share chemical substructure similarities, while others lack a common structure. Examples of similar structures include 2,3-dichlorophenyl piperazine (2,3-DCPP)—denoted in red—which is shared by aripiprazole and cariprazine, while 4-chlorophenyl-4-hydroxypiperidine (CPHP)—denoted in blue—is common for haloperidol and penfluridol. AY9944 and trazodone are examples of chemicals that share similar, though not identical, structures to other DHCR7-inhibiting chemicals.

## Data Availability

Not applicable, as no new data were created for this manuscript.

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
