# Peer review of "Chemical Inhibition of Sterol Biosynthesis"

_biomolecules, 2024, doi:10.3390/biom14040410_

Round 1

Reviewer 1 Report

Comments and Suggestions for Authors

The review manuscript is well-written and well-organized; however, there is one deficit in the current form of this manuscript. Many important scientific descriptions only cite one reference or without citing the reference. Please carefully check this issue throughout the manuscript.

Reviewer 2 Report

Comments and Suggestions for Authors

Very interesting and well-written article. I really like the discussion in which the authors indicate the gaps of knowledge and ask vital questions.

What I missed in the work was more information if there are any clinically relevant side effects linked to the cholesterol synthesis observed after using drugs mentioned in the article (especially psychotropic drugs or beta-blockers, used commonly in therapy).

Furthermore, the purpose of the table 1 is not clear for me. It actually repeats the information on the compounds from figure 1 and the rest of the data is not very informative. In vivo and in vitro assays are mixed and we do not know anything on the study (however, I understand that the authors did not aimed to show the data from those studies but for me the information from the table are useless). The legend, explanation of abbreviations should be under the table, not in the title.

I would also suggest to add a figure showing most important structures from each group of compounds. That would give an idea if there are any similarities in the structures of inhibitors between the groups and within each group.

Good luck!
